# The Combined Effect of Lactic Acid Bacteria and *Galactomyces geotrichum* Fermentation on the Aroma Composition of Sour Whey

**DOI:** 10.3390/molecules28114308

**Published:** 2023-05-24

**Authors:** Kamila Szudera-Kończal, Kamila Myszka, Piotr Kubiak, Natalia Drabińska, Małgorzata Anna Majcher

**Affiliations:** Faculty of Food Science and Nutrition, Poznań University of Life Sciences, 60-624 Poznań, Poland; kamila.szudera-konczal@up.poznan.pl (K.S.-K.); kamila.myszka@up.poznan.pl (K.M.); piotr.kubiak@up.poznan.pl (P.K.); natalia.drabinska@up.poznan.pl (N.D.)

**Keywords:** sour whey, *Galactomyces geotrichum*, LAB, fermentation, aroma-active compounds, aroma biotechnology, stable isotope dilution assay, GC-O, OAV

## Abstract

The increase in demand for food flavorings due to the shortening and simplification of food production technology also entails an increase in the demand for new technologies for their production. The biotechnological production of aromas is a solution characterized by a high efficiency, an independence from environmental factors and a relatively low cost. In this study, the influence of the implementation of lactic acid bacteria pre-fermentation into the production of aroma compounds by *Galactomyces geotrichum* on a sour whey medium on the intensity of the obtained aroma composition was analyzed. The monitoring of the culture in terms of biomass buildup, the concentration of selected compounds, and the pH resulted in the confirmation of interactions between the analyzed microorganisms. The post-fermentation product underwent a comprehensive sensomic analysis for the identification and quantification of the aroma-active compounds. The use of gas chromatography−olfactometry (GC−O) analysis and the calculation of odor activity values (OAVs) allowed 12 key odorants to be identified in the post-fermentation product. The highest OAV was found for phenylacetaldehyde with a honey odor (1815). The following compounds with the highest OAVs were 2,3-butanedione with a buttery aroma (233), phenylacetic acid with a honey aroma (197), 2,3-butanediol with a buttery aroma (103), 2-phenylethanol with a rosy aroma (39), ethyl octanoate with a fruity aroma (15), and ethyl hexanoate with a fruity aroma (14).

## 1. Introduction

Lifestyle changes, combined with the developments in food production technology nutrition in recent years, shape changes in the eating habits of people around the world. Consequently, there is a growing demand for ready-to-eat and ready-to-cook food, as well as ready-to-process food, which shortens the process of preparing meals at home. Another noticeable tendency in the food production market is the simplification and shortening of technological processes. The effect of the changes described above is an increase in the demand for food flavorings, which are necessary to provide a sufficiently intense aroma in processed foods. According to IMARC [1], the global food flavors market reached USD 15.6 billion in 2021, with an estimated value of USD 5 billion higher in 2027.

With the growing demand for food flavorings, there is a need to develop new technologies for their production. Particular emphasis is placed on the development of new technologies for the production of natural-origin flavors, which are currently the most acceptable additives for improving the aroma of food in connection with the growing nutritional awareness of consumers. The growing nutritional awareness of consumers is the result of promoting the healthcare trend. Consequently, food is no longer perceived only in terms of providing basic nutritional properties but also in terms of its impact on vitality. Consumers prefer natural food additives over synthetic ones due to their perception as healthier, the possibility of giving food additional functional properties, and a more sustainable and ecological production process [2,3,4]. The traditional extraction of aroma compounds from natural sources (e.g., plants) may not be sufficient to meet the growing demand for food flavors due to suffering from difficulties, such as seasonality, climate-dependent availability, complexity, time-consuming production technologies, and the resulting relatively high price. Therefore, special attention should be paid to the biotechnological processes of obtaining aroma compounds of a natural origin. These processes include de novo synthesis and the bioconversion of natural-origin precursors by microorganisms or enzymes. Biotechnological processes for the production of aroma compounds are characterized by a higher efficiency, the simplicity of the technology, a high degree of controllability, a care for natural resources, and relatively low production costs, as well as an independence from environmental and sociopolitical factors. Biotechnological processes also offer the possibility of using industrial by-products, making such processes ecological. The main factors limiting the biotechnological production of aroma compounds are related to the possible contamination of the culture. This can be caused by an incomplete sterilization of the medium, a careless implementation of key culture steps or sample handling, and medium composition variability. In addition, in the case of this type of process, it may be necessary to develop technology for the recovery of aroma compounds [2,3,5,6]. However, there is currently a growing interest in aroma mixtures rather than single-aroma compounds in the food industry, which in many cases eliminates the need to develop such technologies. This approach allows for a better reproduction of the natural food aroma, which is usually very complex for any food product [7].

*Galactomyces geotrichum* mold is a natural microflora of dairy products, which affects their nutritional and sensory values [8,9]. *G. geotrichum* is a species referred to as a yeast-like fungus—in the literature it can also be found as *Geotrichum candidum*, an anamorphic form of the old *Galactomyces geotrichum/Geotrichum candidum* complex [10,11]. In our previous studies, we have shown that the strain of *G. geotrichum* isolated from fried cottage cheese has the ability to produce an intense honey–rose–fruit aroma during 7-day cultures on media with sweet whey, sour whey, or buttermilk and a L-phenylalanine supplement [12,13]. The aroma profile of the obtained products resulted mainly in an unusual ratio of phenylacetaldehyde and 2-phenylethanol, which, depending on the culture conditions, is close to or greater than 1: 1. The influence of this phenomenon on the aroma is due to the lower value of the phenylacetaldehyde odor threshold (OT) compared to 2-phenylethanol (60 times lower in water, 10 times lower in oil, and 4 times lower in starch) [14,15]. Moreover, *G. geotrichum* mold is Generally Recognized as Safe (GRAS); therefore, it can be considered in the context of searching for new technologies for enriching food aromas [16].

Lactic acid bacteria (LAB) are part of the microflora of the environment in which the *G. geotrichum* mold naturally occurs. In the fried cheese from which the strain of *G. geotrichum* analyzed in this study was isolated, LAB are responsible for the acidification of milk in the production process [8]. During lactic fermentation, LAB ferment lactose, providing an additional source of carbon for the analyzed molds that are unable to utilize this disaccharide. Moreover, low-molecular-weight metabolites of LAB may be present in the sour whey, which is a component of the culture medium used in this study. It is related to the process of producing sour whey, which is a by-product of the production of cottage cheese [17]. Since the product formed after the fermentation of the medium with sour whey by *G. geotrichum* was characterized by the highest intensity of aroma among all variants analyzed in previous studies, it is assumed that LAB metabolites can affect the metabolism of *G. geotrichum* [12]. In addition to affecting the physicochemical properties of fermented milk products and the metabolic transformation of microbial communities, they also affect their sensory properties, allowing the enrichment of the aroma mixture [18].

The aim of this study was to analyze the combined effect of the inoculation with LAB and *G. geotrichum* on the formation of aroma compounds during the fermentation of the sour whey medium [12]. This was achieved by analyzing the course of the culture and determining changes in the number of microorganisms; the concentration of amino acids, lactic acid, and selected key aroma compounds; and the pH of the medium. The post-fermentation product underwent sensomic analysis including the identification of key odorants by gas chromatography−olfactometry (GC−O), quantitation of the identified compounds using the stable isotope dilution assay (SIDA), and calculation of the aroma activity values (OAVs).

## 2. Results and Discussion

### 2.1. Cultivation Experiments

During the cultivation of LAB and *G. geotrichum* on the medium with sour whey, the pH values, the lactic acid concentration, the amounts of viable cells of analyzed microorganisms, the content of phenylacetaldehyde and 2-phenylethanol, and the amino acid profile were monitored. The monitoring of the culture course was aimed at analyzing the influence of the introduction of LAB pre-fermentation into the process of the production of aroma compositions by *G. geotrichum* on the sour whey medium.

#### 2.1.1. Total Count of LAB and *G*. *geotrichum*, Lactic Acid Concentration, and pH during Bioreactor Culture

To verify the growth of LAB and *G. geotrichum* in one environment, the results of microbiological analyses were compared with the lactic acid concentration and pH during an 8-day culture. Figure 1 shows that both LAB and *G. geotrichum* added after 24 h were characterized by intensive growth on the sour whey medium.

The highest concentration of viable LAB cells was observed on the 2nd day of culture in the bioreactor, and the highest concentration for *G. geotrichum* was observed on the 6th day (4 days from the inoculation of the medium). On the days above, LAB were counted at a level of 11.5 log CFU/mL and *G. geotrichum* were counted at 9.6 log CFU/mL. These dependencies are also observed in the results presented in Figure 2. The first 24 h of the LAB culture resulted in a noticeable decrease in the pH of the medium from 6.9 to 6.1 due to the production of lactic acid by these bacteria. The optimal environment for the growth of *G. geotrichum* is a pH in the range of 5.0–5.5, which was also confirmed in previous studies; therefore, before inoculation with the mold, the pH of the culture was adjusted to 5.0 using lactic acid [12,19]. Within the first 24 h of *G. geotrichum* growth in the presence of LAB, the concentration of lactic acid decreased from 19.3 g/L to 16.0 g/L and then returned to its previous level. The observed decrease in the concentration of lactic acid did not affect the pH. This can be explained by the presence of sodium hydrogen phosphate in the medium, which has buffering properties [20]. In the following days, both the pH and the concentration of lactic acid remained constant. On the 5th day of culture, a decrease in the concentration of lactic acid and an increase in the pH of the medium were observed along with an increase in the concentration of LAB. Therefore, a slight increase in the pH of the environment above 5.0 had a positive effect on the development of LAB. A slight increase in the amount of LAB with the increasing pH of the environment was also observed during the tarhana fermentation with yeast and LAB [21]. With a gradual decrease in the number of LAB and *G. geotrichum* after 6 days of cultivation, a decrease in lactic acid concentration from 18.6 to 17.6 g/L and an increase in pH from 5.0 to 5.4 were observed. The obtained results can be explained by combining the lactic acid production capacity of LAB with the deacidification by the analyzed mold. *G. geotrichum* consume lactic acid and lactate, thus affecting the pH of the environment [22,23]. Lactic acid is not the primary source of carbon in the growth process of the analyzed molds, but it is possible that in the presence of a constant supply of this acid its consumption is intensified to counteract the decrease in pH of the environment. The literature data also reported the possibility of an increase in pH and lactic acid concentration during the cultivation of *G. geotrichum* (in the environment with LAB and separately). These differences may be due to the increase in the total count of certain LAB in co-cultures with *G. geotrichum* and different growth conditions, as well as differences due to the type of *G. geotrichum* strain [11,24,25].

#### 2.1.2. Content of Phenylacetaldehyde and 2-Phenylethanol during Cultivation of LAB and *G. geotrichum*

Phenylacetaldehyde with a honey-like odor and 2-phenylethanol with a rosy-like odor are aroma compounds produced during the transformation of L-phenylalanine in the Ehrlich pathway. This pathway involves the transamination of L-phenylalanine to phenylpyruvate, its decarboxylation to phenylacetaldehyde, and finally its reduction to 2-phenylethanol [26,27]. During the described transformations, both compounds are present in the fermented product, but in most cases a higher concentration of 2-phenylethanol is observed. However, the reverse ratio of these compounds provides a higher aromatizing power of the post-fermentation product due to the 60 times lower odor threshold (OT) of phenylacetaldehyde in water compared to that of 2-phenylethanol [26,27].

When analyzing the effect of LAB growth on the aroma produced by *G. geotrichum*, an experiment was conducted comparing the peak areas of these compounds on each day of the biotechnological process, and the results are presented in Figure 3. It can be seen that, from the detection of 2-phenylethanol in the medium after 3 days of cultivation, the ratio of phenylacetaldehyde to 2-phenylethanol remained higher than 1:1. With the increase in the area of the phenylacetaldehyde peaks with each subsequent day of cultivation, the values found for 2-phenylethanol also increased. The highest values for both compounds were observed on the last day of the biotechnological process.

#### 2.1.3. Amino Acid Concentration during Biotransformation of Sour Whey Medium by LAB and *G. geotrichum*

It is well known that amino acids are some of the precursors of aroma compounds in biotransformation processes [28,29]. To identify the possible precursors of the aroma of the analyzed post-fermentation product, an analysis of the amino acid profile was carried out during the 8-day culture, and the results are presented in Figure 4.

The analyzed medium and the fermentation product contained 11 free amino acids: alanine, glycine, valine, leucine, isoleucine, threonine, γ-aminobutyric acid (GABA), proline, aspartic acid, tyrosine, and L-phenylalanine. L-phenylalanine is a precursor of phenylacetaldehyde and 2-phenylethanol—compounds that were characterized by the highest OAVs in the aroma produced by *G. geotrichum* on substrates with by-products of the dairy industry [12,13,26]. It is an amino acid present in sour whey and is additionally supplemented with the substrate to ensure the possibility of its bioconversion to the highest possible concentration of honey and rose aroma compounds [30]. The results of the amino acid analysis showed that during the fermentation process of the medium with sour whey almost all L-phenylalanine was metabolized, with a radical decrease in its concentration on the last day of cultivation. A decrease in the concentration after 8 days of culture of LAB and *G. geotrichum* on sour whey medium was also observed for leucine, isoleucine, proline, and aspartic acid. Leucine and isoleucine are the precursors of fruity and malty aroma compounds, and the other amino acids may have been involved in cellular metabolism [29,31]. In the case of alanine, glycine, threonine, and tyrosine, an increase in their concentration was observed during cultivation. The concentrations of valine and GABA underwent alternating decreases and increases during the biotechnological process under study. The concentrations of valine and GABA underwent alternating decreases and increases during the biotechnological process under study. In the post-fermentation product, their concentration was finally higher than in the medium before fermentation. The periodic (e.g., in the case of L-phenylalanine) or total increase in the concentration of analyzed amino acids during the 8-day culture can be explained by the complex proteolytic and peptydolytic activity of the analyzed microorganisms, which enables the decomposition of proteins and peptides, enabling further transformations of amino acids [11,32,33].

### 2.2. Sensory Evaluation

The post-fermentation product obtained after culturing LAB and *G. geotrichum* on a medium with sour whey was subjected to sensory evaluation. The results presented in Figure 5 were compared to the odor profile of the product obtained by fermentation of the same medium under the same conditions with only *G. geotrichum* in previous studies [12].

This experiment was aimed at analyzing the influence of introducing the pre-fermentation of the sour whey medium with LAB into the analyzed biotechnological process on the aroma profile of the obtained aroma composition. The product obtained after the fermentation of sour whey substrate by LAB and *G. geotrichum* was characterized by a high intensity of honey (9.8), rose (7.1), butter (7.1), fruit (6.9), and caramel (5.4) aroma notes.

The panelists pointed out a significant increase in the intensity of the buttery aroma (by 3.9) and noticeably more intense caramel (by 2.2) and fruity (by 1.1) aroma notes after introducing the initial fermentation with LAB to the analyzed biotechnological process. The honey aroma was again assessed as very strong, with a slight increase in perceptibility (by 0.3). In the case of the rose aroma, a decrease in its intensity (by 1.6) was observed in relation to the aroma produced by *G. geotrichum*, though it still remained a strongly perceptible note of the analyzed aroma. Cheese notes (2.2) and sour notes (2.6) were the least perceptible. In relation to the aroma composition obtained after the fermentation of sour whey exclusively by *G. geotrichum*, there was a decrease in cheese note intensity by 1.9.

### 2.3. Identification of Key Aroma Compounds in a Post-Fermentation Product Using GC−O Analysis and Calculation of OAVs

Samples collected after the bioreactor culture of LAB and *G. geotrichum* on the sour whey medium were subjected to SAFE extraction to perform a full characterization of the obtained aroma composition. During the GC-O analysis of the post-fermentation product, 18 aroma compounds were identified. The results of the quantification of the identified compounds and the OAVs are presented in Table 1.

An important element of the interpretation of the results of the identified aroma compounds in the analyzed composition are the OAVs. These are indicators calculated by dividing the concentration of a given compound by its OT value, determining the aroma compound activity. By comparing the OAVs of all identified aroma compounds in a given product, it is possible to determine their contribution to the final impression of the total aroma [35]. The OAVs calculated for aroma compounds in the product obtained after fermentation with LAB and *G. geotrichum* indicate that 12 of the 18 identified compounds have OAVs greater than 1.0, potentially contributing to the overall aroma of the fermentation product. In the analyzed aroma composition, five compounds were identified as having OAVs ranging from 0 to 1, five ranging from greater than 1 to 10, three ranging from greater than 10 to 100, three ranging from greater than 100 to 500, and one compound having OAV greater than 500. In the case of dimethyl sulfone, the OAV could not be calculated due to the unavailability of its OT value. Methyl decanoate, however, was identified at the stage of GC-O analysis, but the concentration of this compound turned out to be below the detection limit, which also made it impossible to calculate its OAV.

The obtained results indicate that the key odor compound shaping the aroma of the sour whey post-fermentation product with the highest OAV is phenylacetaldehyde (1815). At the same time, this compound was not characterized by the highest concentration among all identified. Such a high OAV is related to the very low OT (in water) value of phenylacetaldehyde, which is 4 µg/kg [34]. It should be noted that this value is 1.65 times lower than in the case of the aroma composition produced as a result of the fermentation of the sour whey medium exclusively by *G. geotrichum*; however, it did not affect the intensity of the honey aroma in the analyzed product. Moreover, the intensity of the honey note of the aroma increased by 0.3 [12]. This phenomenon is related to the perception of aroma. Although each aroma compound is characterized by a specific odor, the loss of their individual aroma descriptors is already observed in mixtures of four compounds. The compounds present in the mixture may have a synergistic effect and completely lose their individual fragrance notes, as well as condition the appearance of an aroma note unrelated to any of them. A complex aroma consisting of more than four aroma compounds (which is the case in most food products) should be considered not only as the sum of the aroma of individual compounds but mainly as a total effect [7,36,37]. Phenylacetaldehyde is formed from phenylalanine during the Ehrlich pathway. During bioconversion, phenylacetaldehyde can be transformed into 2-phenylethanol with a rosy aroma or phenylacetic acid with a honey aroma, which was also identified in the post-fermentation product with OAVs 39 and 197, respectively [26,29,38]. Phenylacetaldehyde, 2-phenyl ethanol, and phenylacetic acid were found in the tested variant in a ratio of 1.7:1:2.5. Comparing this result to previous studies with *G. geotrichum* without pre-fermentation with LAB, where the ratio was 1.7:1:0.2, a significant increase in the concentration of phenylacetic acid is observed while maintaining the current ratio of phenylacetaldehyde to 2-phenylethanol, which ensures an intense honey–rose aroma [12]. Similar observations were noted in Whetstine’s research [36]: they concluded that phenylacetaldehyde combined with phenylacetic acid provides a more intense aroma in Cheddar cheese. Thus, a high concentration of phenylacetic acid may most likely significantly increase the intensity of the honey aroma of the final fermentation mixture.

In addition to compounds with a honey–rose aroma, the key odorants were also those with a buttery odor, which can also be observed in the aroma profile of the analyzed post-fermentation product. 2,3-butanedione was the second compound with the highest OAV (233), and 2,3-butanediol was fourth (103). The resulting concentration of 2,3-butanedione in the analyzed aroma composition is as much as 6.4 times higher than in previous studies involving *G. geotrichum* fermenting the sour whey medium. This correlates very clearly with sensory analysis. In combination with the identification of 2,3-butanediol at a concentration of 15,464 µg/kg, this resulted in a significant increase in the intensity of the pleasant buttery aroma in the analyzed composition. The presence of these compounds in such concentrations was undoubtedly the effect of LAB, which are characterized by the ability to produce them. Compounds with a buttery aroma are some of the compounds shaping the aroma of yoghurt and other dairy products [33]. 2,3-butanedione is a compound formed during the transformation of sugars and lipids via various bacteria and has been reported as a key odorant in cheeses, such as fried cottage cheese, Cheddar, Camembert, and Lazur [8,36,39,40]. 2,3-butanediol, which is a reduced form of acetoin (also produced by LAB), has been identified as one of the key odorants in the traditional fermented milk product called “Lben” [41].

Among the compounds identified in the aroma mixture obtained after the fermentation of the sour whey medium by LAB and *G. geotrichum*, six compounds with a fruity aroma note were found. One of these compounds is 3-methyl-1-butanol, which is generated in the biotransformation of leucine [7,29]. During cultivation, a decrease in the concentration of this amino acid was observed, indicating a pathway for generating this aroma compound. 3-Methyl-1-butanol is responsible for the fruity smell of many fermented products, such as wine, cider, fermented meat products, and pumpernickel sourdough [42,43,44,45]. The concentration of this compound in the analyzed fermentation product was 95 times lower than in the product after the fermentation of the sour whey medium only by *G. geotrichum* [12]. Nevertheless, an increase in the intensity of the fruity aroma was observed in the composition after fermentation by LAB and *G. geotrichum*. This draws attention to the possibility of intensifying the fruity aroma by combining various esters and 3-methyl-1-butanol in one mixture. Among the identified esters, the highest OAVs were found in ethyl octanoate (15) and ethyl hexanoate (14), which are also some of the eight esters included in the aroma of yoghurt as a result of fermentation by LAB and key odorants in alcoholic beverages [7,33].

Another key odorant is vanillin with an OAV of 1.1, which has not yet been found in the aroma produced by *G. geotrichum*. Vanillin, widely used in the food and cosmetic flavor industry due to its pleasant vanilla aroma, can be formed biotechnologically by some *Lactobacillus* sp. by the transformation of ferulic acid [28,46]. Four acids were also identified in the fermentation product, of which only octanoic acid with a musty odor was characterized by an OAV > 1. It is worth noting that a 35 and 50 times decrease in the concentration of acetic acid and butanoic acid was observed, respectively, in relation to the culture without LAB pre-fermentation [12]. The decrease in the concentration of these acids reduces unpleasant odors, such as vinegar-like and cheese-like odors, which result from their high concentrations in the product. The last key odorant in the product resulting from the fermentation of the sour whey medium by LAB and *G. geotrichum* is dimethyl sulfone. This compound was found in the analyzed product in the highest concentration (53,449 µg/kg) among all identified compounds, but no sulfur note of aroma was found for it. The possibility of its synthesis by biotechnology in food products has not been demonstrated so far, but it has been described as a new marine biogenic emission. Therefore, the most probable pathway of dimethyl sulfone production is the described oxidation of dimethyl sulfide by OH radicals [47]. Dimethyl sulfone should be considered in terms of the element of the aroma composition affecting its overall perception.

It should be noted that four of the identified aroma compounds (2,3-butanedione, acetic acid, butanoic acid, and vanillin) belong to the group referred to as “generalists”, constituting the key aroma compounds of more than 25% of food products, while seven compounds (phenylacetaldehyde, 2-phenylethanol, phenylacetic acid, ethyl hexanoate, ethyl octanoate, hexanoic acid, and 3-methyl-1-butanol) are included in the “intermediaries” group, shaping the aroma of from 5 to 25% of food products [7].

The influence of mold *G. geotrichum* on LAB is described in the literature. In Chaves-López’s [11] research, it was observed that *G. geotrichum*, when present in an environment with LAB, increased their viability. This has the effect of compensating for the decrease in the lactic acid production capacity of the LAB with the duration of the culture. The obtained results show that the interactions between *G. geotrichum* and LAB are mutual. The implementation of LAB pre-fermentation to the production of aroma compounds by *G. geotrichum* on a sour whey medium increased the amount of aroma-active odorants identified in the obtained mixture. Changing the composition of the fermentation product resulted in an increase in the intensity of pleasant fragrance notes (honey, rosy, caramel) and a reduction in irritating odors by their lower concentration.

## 3. Materials and Methods

### 3.1. Chemicals

Spray-dried sour whey was obtained from Laktopol (Suwałki, Poland). L-Phenylalanine, lactic acid, sodium sulfate, dichloromethane, diethyl ether, and Man Rogosa Sharpe (MRS) agar were purchased from Sigma-Aldrich (Poznań, Poland). Yeast extract and a medium with chloramphenicol were obtained from BTL (Łódź, Poland). Citric acid, Na_2_HPO_4_·2H_2_O, and MgCl_2_ were obtained from POCH (Gliwice, Poland). Inulin was purchased from Hortimex Plus (Konin, Poland). An EZ: Faast™ Kit for Free (Physiological) Amino Acids was obtained from Phenomenex (Aschaffenburg, Germany). The following reference aroma compounds were purchased from Sigma-Aldrich (Poznań, Poland): 2,3-butanedione, acetic acid, 3-methyl-1-butanol, methyl butanoate, 2,3-butanediol, butanoic acid, methyl hexanoate, dimethyl sulfone, ethyl hexanoate, hexanoic acid, phenylacetaldehyde, 2-phenylethanol, methyl octanoate, ethyl octanoate, phenylacetic acid, octanoic acid, methyl decanoate, vanillin, and lactic acid. The following stable isotopes were obtained from AromaLAB (Freising, Germany): [^13^C_4_] 2,3-butanedione, [^13^C_1_] acetic acid, [^2^H_2_] 3-methyl-1-butanol, [^2^H_9_] ethyl 3-methylbutanoate, [^2^H_5_] phenylacetaldehyde, [^2^H_5_] 2-phenylethanol, [^2^H_3_] vanillin, and [^2^H_8_] naphthalene.

### 3.2. Microorganisms

The strain 32 *G. geotrichum* came from the collection of the Food Volatilomics and Sensomics Group, Poznań University of Life Sciences. The analyzed mold strain was isolated from Wielkopolski fried cottage cheese produced in the vicinity of Poznań in Poland during the ripening stage. The *G. geotrichum* strain was protected by freeze-drying. The preparation of the *G. geotrichum* strain for this study followed the methodology described in our previous studies [12,13].

A lyophilized starter culture of heterofermentative LAB (*Lactococcus lactis* subsp. *lactis*, *Lactococcus lactis* subsp. *cremoris*, *Lactococcus lactis* subsp. *lactis* biovar *diacetilactis*, *Leuconostoc mesenteroides* subsp. *cremoris*) for the production of cottage cheese was obtained from Biochem Srl (Montelibretti, Italy).

### 3.3. Bioreactor Culture

The cultivation of LAB and *G. geotrichum* was carried out in a 2.3 L Labfors 5 bioreactor (Infors HT, Bottmingen, Switzerland) with a working volume of 2 L. The microorganisms were grown in a medium containing, per liter (modified, based on Grygier et al. [48]), 22.8 g of Na_2_HPO_4_·2H_2_O, 0.5 g of MgCl_2_, 0.17 g of yeast extract, 60 g of sucrose, 130 g of spray-dried sour whey, and 21 g of L-phenylalanine. Sucrose, sour whey, and L-phenylalanine were added to the sterilized medium after exposure to UV radiation for 30 min due to the risk of changes in the aroma profile of the post-fermentation product due to the effect of high temperatures on these compounds. The medium was inoculated with a cottage cheese starter culture according to the producer’s recommendations (6 × 10^9^ CFU) to best reflect the natural growth environment of *G. geotrichum* isolated from fried cottage cheese. The pre-fermentation of the sour whey medium by LAB was carried out for 24 h at 30 °C with stirring at a speed of 150 rpm. After initial fermentation, the pH of the medium was adjusted to 5.0 with lactic acid and inoculated with 3.4 × 10^7^ CFU of lyophilized *G. geotrichum* mold. The cultivation was carried out for 7 days at 30 °C with stirring at 150 rpm and aeration to 1.5 vvm, with the inlet air sterilized by filtration. The pH value was not adjusted during the culture but was constantly monitored. A blank test was also performed using the same culture conditions without the inoculation of sour whey medium with LAB and *G. geotrichum.*

### 3.4. Cultivation Experiments

#### 3.4.1. Total Count of LAB and *G. geotrichum*

In order to determine the number of *G. geotrichum* colony-forming units (CFUs), serial dilutions of the culture were prepared and plated onto plates with a solid medium with chloramphenicol, which were incubated for 72 h at 30 °C under aerobic conditions. The determination of the number of LAB CFUs was carried out in an analogous way, using the MRS agar and incubation for 48 h at 30 °C under anaerobic conditions. After the incubation of the agar plates, the CFUs were counted and the results were converted to CFUs per milliliter. The experiment was performed on each culture day in triplicate.

#### 3.4.2. Determination of Lactic Acid Concentration

Samples taken during bioreactor culture of LAB and *G. geotrichum* were diluted with distilled water and centrifuged for 10 min at 13,000× *g* using a Biofuge Primo R centrifuge (Heraeus, Hanau, Germany). The supernatant obtained was passed through 0.45 µm mixed cellulose ester syringe filters (Lab Logistics Group GmbH, Meckenheim, Germany). The obtained samples underwent HPLC analysis to identify and quantify the lactic acid, using an Agilent 1200 series chromatograph with an autosampler and refractive index detector (RID) (Agilent Technologies, Santa Clara, CA, USA). The apparatus was equipped with a REZEX-ROA column (300 mm × 7.8 mm, Phenomenex, Aschaffenburg, Germany). The samples (10 µL) were injected into the HPLC column using the splitless mode. Lactic acid was analyzed at 40 °C with 0.005 N H_2_SO_4_ as a mobile phase at a flow rate of 0.6 mL/min. Lactic acid was identified by comparison with a reference compound based on the retention index. Quantitation was performed with an external standard curve. The analysis was carried out on samples taken during each day of culture in the bioreactor in order to study changes in lactic acid concentration during its duration.

#### 3.4.3. Analysis of the Amino Acid Profile

Samples taken during the cultivation of LAB and *G. geotrichum* on a whey medium were diluted 1:1 with distilled water and centrifuged at 4000 rpm for 15 min using a MPW-223e centrifuge (MPW MED. INSTRUMENTS, Warsaw Poland). The supernatant obtained was passed through 0.45 µm cellulose acetate syringe filters (Lab Logistics Group GmbH, Meckenheim, Germany). The sample obtained was directly analyzed using the EZ: Faast™ Kit for Free (Physiological) Amino Acids (Phenomenex, Aschaffenburg, Germany) according to the producer’s recommendations. The analytical procedure involves the solid-phase extraction of 50 μL of extract, followed by derivatization and liquid–liquid extraction. Amino acids were separated in the Agilent 7890A gas chromatograph (Agilent Technologies, Santa Clara, CA, USA) equipped with an autosampler G45134, using the Agilent 5975C Mass Selective Detector. Derivatized amino acids were separated in the ZB-AAA EZ Faast™ capillary column (10 m × 0.25 mm, Phenomenex, Aschaffenburg, Germany). The carrier gas was helium (1.5 mL/min). The samples (2 µL) were injected in the split mode set as 1:50 and 1:5 for phenylalanine and other amino acids, respectively. The oven temperature was initially set at 110 °C and then increased to 320 °C (30 °C/min). The injector and ion source temperatures were 250 °C and 240 °C, respectively. Mass spectra were obtained by electron ionization (EI) over the range of 35–550 *m*/*z*. The electronic impact energy was 70 eV. The amino acids were identified and quantified using standards for each amino acid and normalized in relation to the internal standard (norvaline). The analysis was carried out on samples taken during each day of the bioreactor culture in order to examine changes in the amino acid profile during its duration.

### 3.5. Sensory Evaluation

The sour whey medium after fermentation with LAB and *G. geotrichum* was subjected to descriptive sensory evaluation by ten experienced panelists. In this experiment, seven aroma descriptors (honey, caramel, rosy, fruity, cheesy, sour, and buttery) were assessed in triplicate during separate profile sessions. The analyzed characteristics were selected from the basic flavor descriptive language (Givaudan Roure Flavor [49]) and applied in previous studies. Sensory evaluation was performed by presenting to the panelists a 20 g sample of the post-fermentation product in 100 mL glass containers at room temperature. Panelists scored each odor descriptor on a 10 cm linear scale, with the beginning labeled “none” and the end labeled “very strong”. The results have been converted to numerical values. The data obtained for each descriptor were averaged and subjected to t-test analysis.

### 3.6. Extraction of Odor-Active Compounds Using the Solvent-Assisted Flavor Evaporation (SAFE) Method

The aroma mixture obtained after the bioreactor culture of LAB and *G. geotrichum* on sour whey medium underwent extraction using the SAFE method described by Engel et al. [50] in order to isolate the odor-active compounds. The samples (50 g) were mixed with dichloromethane (100 mL) and spiked with the internal standard [^2^H_8_] naphthalene (25 µg), then they were shaken for 2 h on a horizontal shaker. After the extraction process, the samples were subjected to the isolation of volatile compounds using the SAFE method. The obtained extracts were dried over anhydrous sodium sulfate and concentrated using a Kuderna Danish concentrator (Sigma-Aldrich, Poznań, Poland) to a volume of approximately 500 µL.

### 3.7. GC-O and GC-GC Analysis

#### 3.7.1. Gas Chromatography–Olfactometry (GC-O)

The identification of odor-active compounds was carried out by subjecting extracts obtained by the SAFE method to GC-O analysis. The identification was performed using an HP 5890 chromatograph (Hewlett- Packard, Wilmington, DE, USA) with two columns of different polarities: SPB-5 (30 m × 0.53 mm × 1.5 μm) and SUPELCOWAX 10 (30 m × 0.53 mm × 1 μm) (Supelco, Bellefonte, PA, USA). The effluent was divided between the olfactometry port with humidified air as a makeup gas and a flame ionization port using a Y splitter in GC. The operating conditions for the SPB-5 column were as follows: an initial oven temperature of 40 °C (1 min) raised at 6 °C/min to 180 °C and at 20 °C/min to 280 °C. For the SUPELCOWAX 10 column, the operating conditions were as follows: an initial oven temperature of 40 °C (2 min) raised to 240 °C at a 6 °C/min rate and held for 2 min isothermally. The injection of the SAFE extract (2 μL) into a GC column was performed in a splitless mode. In order to detect active regions and specific notes of selected volatiles, three panelists sniffed the GC-O effluent. For all peaks and flavor descriptors with specific retention times, retention indices were calculated in order to compare the results with those obtained by GC−MS and literature data. For each compound, the retention indices (RIs) were calculated using a homologous series of C7−C24 n-alkanes.

#### 3.7.2. Comprehensive Two-Dimensional Gas Chromatography (GC-GC)

For the identification and quantitation of the aroma compounds, samples were run on a comprehensive gas chromatography–mass spectrometry system—GCxGC-ToF-MS (Pegasus 4, LECO, St. Joseph, MI). The GC was equipped with an SLB-5MS column (30 m × 0.25 mm × 0.25 μm) and SPB-50 (1.2 m × 0.25 mm × 0.25 μm) as a second column. For two-dimensional analysis modulation (liquid N modulator by ZOEX), the time was optimized and set at 4 s, and mass spectra were collected at a rate of 150 scans/s. The transfer line was heated up to 260 °C, and the ion source was heated to 220 °C, respectively. For all the volatiles, identification was performed by a comparison of mass spectra, retention indices (RI) on two columns of different polarities to those of standard compounds, the National Institute of Standards and Technology (NIST) 09 Mass Spectral Library, and literature data.

### 3.8. Quantitation by Stable Isotope Dilution Assays (SIDAs)

For all of the compounds identified in the GC-O experiment, stock internal standards of the labeled isotopes were prepared in diethyl ether and added to the analyzed samples in a concentration similar to that of the relevant analyte. The volatiles were analyzed by GCxGC-ToF-MS monitoring the intensities of the respective ions given in Table 2. For all the volatiles, response factors were calculated in the standard mixture of labeled and unlabeled compounds at a known concentration of 500 ppb. The concentrations in the samples were calculated from the peak area of the analyte and its corresponding internal labeled standard obtained for selected ions. Finally, odor activity values (OAVs) were calculated by dividing the concentration of a given analyte by its odor threshold (OT) determined in water.

## 4. Conclusions

The results of this study indicated that in addition to the known effect of *G. geotrichum* on LAB in one environment the dependence is also reversed. LAB are part of the microflora of the environment where the mold *G. geotrichum* is naturally present. The initial fermentation of sour whey by LAB reproduces the natural conditions for the development of the analyzed molds by the presence of the effects of their metabolism. LAB’s ability to produce lactic acid adjusted the pH of the medium to the growth of *G. geotrichum*, reducing the amount of lactic acid added for its regulation. Moreover, lactic acid synthesized throughout the culture was an additional source of carbon for the analyzed mold. Low molecular weight metabolites resulting from LAB growth can affect *G. geotrichum* by imitating their natural development environment.

The combined effect of the LAB and *G. geotrichum* fermentation of sour whey increased the amount of odor-active compounds in the aroma composition. The GC-O identified 18 aroma compounds. The OAV calculation allowed for the identification of 12 aroma-active compounds among them. The key odorant shaping the aroma of the analyzed composition is phenylacetaldehyde with a honey aroma note. The next compounds with the highest OAVs were 2,3-butanedione (buttery), phenylacetic acid (honey), 2,3-butanediol (buttery), 2-phenylethanol (rosy), ethyl octanoate (fruity), and ethyl hexanoate (fruity). The introduction of LAB pre-fermentation into the bioconversion process by *G. geotrichum* also resulted in changes in the concentration of aroma compounds identified in recent studies. As a result, the intensity of pleasant aroma notes (honey, buttery, rosy, caramel) increased and the irritating aroma notes of acids (vinegar-like, cheesy) were reduced.

The obtained results provide basic knowledge on the possibility of intensifying the aroma produced by *G. geotrichum* mold. The effect of the combined fermentation of sour whey by LAB and *G. geotrichum* provides an aroma composition with an intense, honey–buttery aroma. It gives the opportunity to enrich the aroma of not only dairy industry products but also, for example, the products of the bakery, confectionery, and brewing industries.

## Figures and Tables

**Figure 1 molecules-28-04308-f001:**
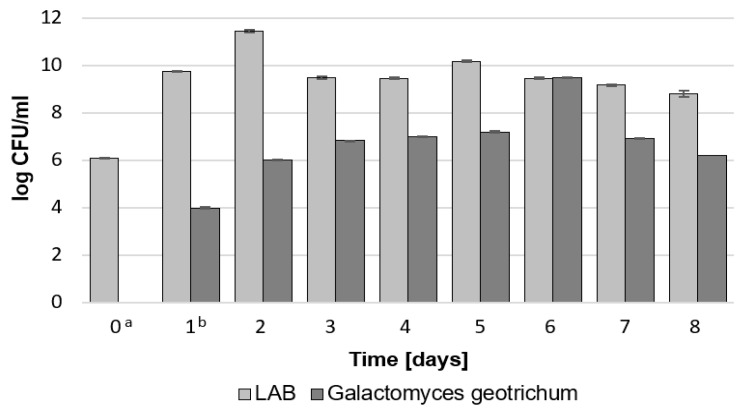
The concentration of LAB and *G. geotrichum* during 8-day culture on sour whey medium. ^a^ LAB inoculation. ^b^
*G. geotrichum* inoculation. Error bars represent ± SD from the replicates.

**Figure 2 molecules-28-04308-f002:**
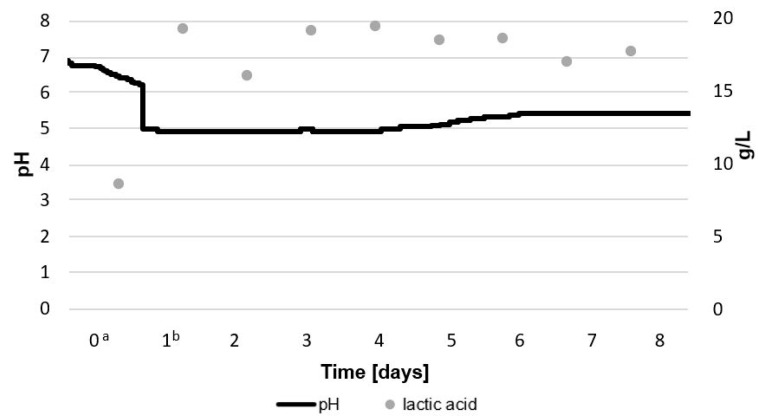
Lactic acid concentration and pH during an 8-day culture of LAB and *G. geotrichum* on acid whey medium. Lactic acid concentration was determined for samples taken every 24 h and pH was monitored every minute. ^a^ LAB inoculation. ^b^ *G. geotrichum* inoculation.

**Figure 3 molecules-28-04308-f003:**
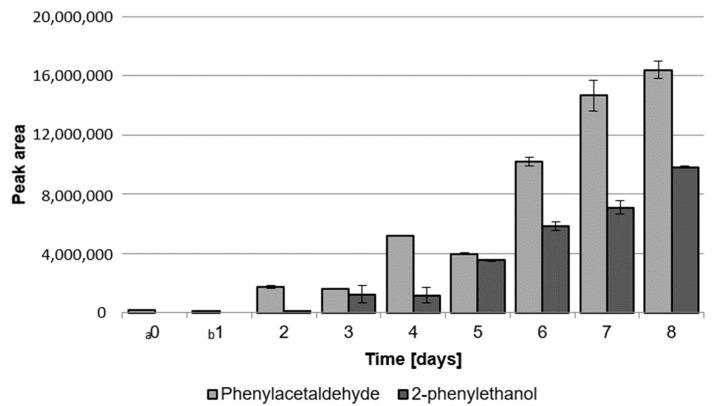
Peak areas of phenylacetaldehyde and 2-phenylethanol extracted from post-fermentation products obtained by biotransformation of sour whey medium by LAB and *G. geotrichum*. ^a^ LAB inoculation. ^b^ *G. geotrichum* inoculation. Error bars represent ± SD from the replicates.

**Figure 4 molecules-28-04308-f004:**
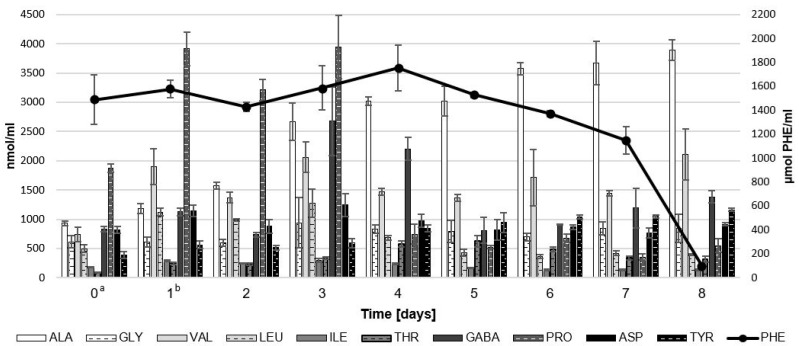
The concentration of free amino acids during 8-day fermentation of sour whey medium by LAB and *G. geotrichum*. ^a^ LAB inoculation. ^b^ *G. geotrichum* inoculation. Error bars represent ± SD from the replicates.

**Figure 5 molecules-28-04308-f005:**
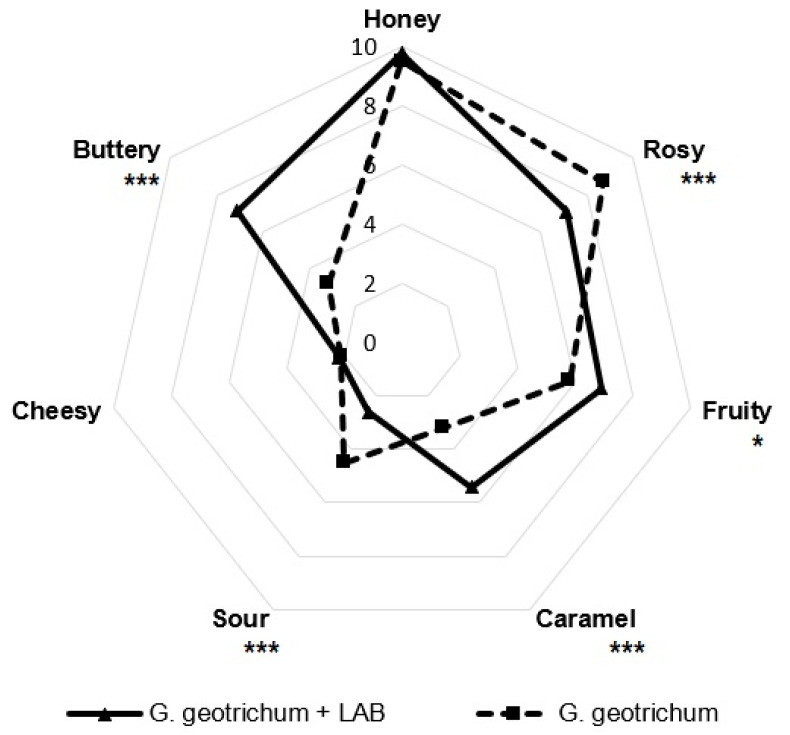
Sensory profiles of flavor mixtures after fermentation with LAB and *G. geotrichum* and only with *G. geotrichum* [12]. * *p* < 0.05; *** *p* < 0.001.

**Table 1 molecules-28-04308-t001:** Aroma-active compounds identified in post-fermentation products from sour whey.

Compound ^a^	Odor ^b^	RI-DB 5 ^c^	RI-Wax ^c^	Concentration (µg/kg) ^d^	OT in Water (µg/kg) ^e^	OAV ^f^
2,3-butanedione	buttery	670	991	3500	15	233
acetic acid	vinegar	691	1445	7080	99,000	<1
3-methyl-1-butanol	fruity	719	1204	1269	980	1.3
methyl butanoate	fruity	734	1052	375	60	6.2
2,3-butanediol	buttery	810	1583	15,464	150	103
butanoic acid	cheesy	836	1620	367	1000	<1
methyl hexanoate	fruity	923	1245	164	90	1.8
dimethyl sulfone	sulfuric	959	-	53,449 *	n.d.	-
ethyl hexanoate	pineapple	988	1229	17	1.2	14
hexanoic acid	sweaty	993	1224	2823	3000	<1
phenylacetaldehyde	honey	1080	1644	9440	5.2	1815
2-phenylethanol	rosy	1124	1920	5400	140	39
methyl octanoate	fruity	1131	1378	147	200	<1
ethyl octanoate	fruity	1194	1441	132	8.7	15
phenylacetic acid	honey	1260	2568	13,400	68	197
octanoic acid	musty	1268	1420	554	190	2.9
methyl decanoate	sweaty	1325	1590	<LOD ^g^	4	-
vanillin	vanilla	1406	2590	56	53	1.1

^a^ The compounds identified by comparison with reference compounds based on the following criteria: retention index (RI), mass spectra obtained by MS (EI), and odor quality at the sniffing port. ^b^ Odor perceived at the sniffing port. ^c^ Retention indices on DB-5 and SUPELCOWAX 10 columns. ^d^ Mean values based on three replicates with RSD value ≤ 11%. ^e^ OT: odor thresholds in water [34]. ^f^ OAV: odor activity values calculated by dividing the concentration of an analyte by its odor threshold value in water. ^g^ LOD—limit of detection. * Concentration estimated based on the internal standard [^2^H_8_] naphthalene.

**Table 2 molecules-28-04308-t002:** Labeled standards and quantitation ions used for SIDA (stable isotope dilution assay) concentration calculations of 13 key odorants present in the post-fermentation products.

Compound ^a^	Quant. Ions *(m*/*z)* ^b^ (*m*/*z*) ^b^	Labeled Standards ^c^	Ion IS (*m*/*z*) ^d^
2,3-butanedione	86	[^13^C_4_] 2,3-butanedione	90
acetic acid	60	[^2^H_3_] acetic acid	63
3-methyl-1-butanol	70	[^2^H_2_] 3-methyl-1-butanol	72
methyl butanoate	102	[^2^H_9_] ethyl 3-methylbutanoate	139
2,3-butanediol	90	[^13^C_4_] 2,3-butanedione	90
butanoic acid	73	[^2^H_7_] butanoic acid	77
methyl hexanoate	130	[^2^H_9_] ethyl 3-methylbutanoate	139
dimethyl sulfone	94	[^2^H_8_] naphthalene	131
ethyl hexanoate	144	[^2^H_9_] ethyl 3-methylbutanoate	139
hexanoic acid	116	[^2^H_7_] butanoic acid	77
phenylacetaldehyde	120	[^2^H_5_] phenylacetaldehyde	125
2-phenylethanol	122	[^2^H_5_] 2-phenylethanol	127
methyl octanoate	158	[^2^H_9_] ethyl 3-methylbutanoate	139
ethyl octanoate	172	[^2^H_9_] ethyl 3-methylbutanoate	139
phenylacetic acid	136	[^2^H_5_] 2-phenylethanol	127
octanoic acid	122	[^2^H_7_] butanoic acid	77
methyl decanoate	186	[^2^H_9_] ethyl 3-methylbutanoate	139
vanillin	152	[^2^H_3_] vanillin	155

^a^ Quantified compounds; ^b^ ions used for quantitation of analytes; ^c^ corresponding labeled standards used for quantitation; ^d^ ions of internal standards (labeled isotopes) used for quantitation.

## Data Availability

The data presented in this study are available on request from the corresponding authors.

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
