# Peer review of "The Combined Effect of Lactic Acid Bacteria and *Galactomyces geotrichum* Fermentation on the Aroma Composition of Sour Whey"

_molecules, 2023, doi:10.3390/molecules28114308_

Round 1

Reviewer 1 Report

This manuscript evaluated the combined effect of LAB and Galactomyces geotrichum fermentation of sour whey on the aroma composition, it will provides the unique honey-buttery aroma. Overall this is a very interesting work and shows some novelty.

However, This study did not consider whether the concentration ratio of LAB and Galactomyces geotrichum would have different effects on co-fermentation. A supplementation study is needed to inoculate different concentration ratios of lactic acid bacteria and Galactomyces geotrichum on the culture medium and provide the optimal ratio of lactic acid bacteria and Galactomyces geotrichum for fermentation to produce honey-buttery aroma compounds.

Some other revisions are also needed:

1. The English grammar of whole article should be improved and corrected.

2. Title: Please change the title “Combined effect of LAB and Galactomyces geotrichum fermentation of sour whey on the aroma composition.” to “Combined effect of Lactic Acid Bacteria and Galactomyces geotrichum fermentation on the aroma composition of sour whey”.

3. Give the full name of SIDA instead of abbreviation, as itl appears for the first time in the Keywords.

Introduction:

4. Line 37-40 Why the natural-origin flavors are the most acceptable additives currently? Some reasons and advantages of them should be given. What is the relationship between the use of natural-origin flavors and the increase in consumers' nutritional awareness? Please explain in detail.

5. Line 85-86 How the LAB influences the metabolism of G. geotrichum. The influence mechanism of metabolites and properties of LAB on G. geotrichum should be given to explain this assumption.

Results and Discussion:

6. The discussion section should be in-depth and comprehensive, especially comparative analysis with some latest literature to show the novelty of this study.

7. Line 125: 19.3 g/L instead of 19.3 g/l

8. Line 125: Please discuss the reasons of the decrease after the second day, and why the LAB concentration increase again on the fifth day?

9. Line 140-142 Figure 2: The position of the third grey dot in Figure 2 is significantly lower than the second one and the fourth one, so why the pH value is constant between these three dots?

10. Line 142, every day not every minute.

11. From sensory evaluation, there is still a sour flavor in LAB and G. geotrichum. Sour is a pungent taste, and it may influence the honey-buttery aroma. The extraction method for honey-buttery aroma or method for removing sour substances should be mentioned.

Materials and Methods:

12. Line 384: The LAB concentration should be mentioned during the pre-fermentation.

Conclusions:

13. Line 531-533: Not all kinds of food are suitable for this honey-buttery aroma. Please indicate some specific fields of the food industry which can benefit from this study. The conclusion is not considerate and detailed. Please provide more suggestions.

The English grammar of whole article should be improved and corrected.

Author Response

file attached

Reviewer 2 Report

The authors are well acquainted with the subject of Galactomyces geotrichum, and have already published several papers on the subject. A new approach is co-fermentation with the participation of LAB.

However, I think that the authors should supplement or correct some imperfections before accepting them for publication.

Results and Discussion:

Fig. 3 . Why authors present the results as peak areas of phenylacetaldehyde and 2-phenylethanol. If they are not able to convert this to concentrations of compounds, they should at least provide a different description on the y-axis.

Materials and Methods

l.349 "and medium with chloramphenicol" Please explain what kind of medium for which microorganism you are talking about?

l. 395  "agar plates with chloramphenicol". Please describe this medium more precisely

l.404 "cellulose mixed cellulose ester syringe filters" please check

Conclusions are too long. 

l. 524-526

The introduction of pre-fermentation with LAB into the aroma bioconversion process The introduction of LAB pre-fermentation into the bioconversion process by G. geotrichum also  resulted in changes in the concentration of aroma compounds identified in recent studies" please check

in the list of literature there are incorrectly spelled names of microorganisms, please correct it

Author Response

file attached

Round 2

Reviewer 1 Report

This study did not consider whether the concentration ratio of LAB and Galactomyces geotrichum would have different effects on co-fermentation. A supplementation study is needed to inoculate different concentration ratios of lactic acid bacteria and Galactomyces geotrichum on the culture medium and provide the optimal ratio of lactic acid bacteria and Galactomyces geotrichum for fermentation to produce honey-buttery aroma compounds.

 Moderate editing of English language required.